# UltraViolet SANitizing System for Sterilization of Ambulances Fleets and for Real-Time Monitoring of Their Sterilization Level

**DOI:** 10.3390/ijerph19010331

**Published:** 2021-12-29

**Authors:** Zuleika Michelini, Chiara Mazzei, Fabio Magurano, Melissa Baggieri, Antonella Marchi, Mauro Andreotti, Andrea Cara, Alessandro Gaudino, Marco Mazzalupi, Francesca Antonelli, Lorenzo Sommella, Silvia Angeletti, Elena Razzano, Arnaud Runge, Paolo Petrinca

**Affiliations:** 1National Center Global Health, Istituto Superiore di Sanità, 00161 Rome, Italy; chiara.mazzei@iss.it (C.M.); mauro.andreotti@iss.it (M.A.); andrea.cara@iss.it (A.C.); 2Department of Infectious Diseases, Istituto Superiore di Sanità, 00161 Rome, Italy; fabio.magurano@iss.it (F.M.); melissa.baggieri@iss.it (M.B.); antonella.marchi@iss.it (A.M.); 3Department of Medicine and Surgery, University of Perugia, 06123 Perugia, Italy; alessandro.gaudino@unipg.it; 4OMICA, 00143 Rome, Italy; das@omica.it (M.M.); admin@omica.it (F.A.); paolo.petrinca@omica.it (P.P.); 5Department of Clinical Pathology, Università Campus Bio-Medico di Roma, 00128 Rome, Italy; l.sommella@unicampus.it (L.S.); s.angeletti@unicampus.it (S.A.); 6European Space Agency—ECSAT, Harwell Campus, Didcot OX11 0FD, UK; Elena.Razzano@ext.esa.int; 7European Space Agency—ESTEC, 2201 AZ Noordwijk, The Netherlands; Arnaud.Runge@esa.int

**Keywords:** UVC, SARS-CoV-2, MDR bacteria, Sanitizing System, ozone, Satellite System

## Abstract

Background: The contamination of ambulances with pathogenic agents represents a potential threat for the public health, not only for common pathogens but also for severe acute respiratory syndrome coronavirus 2 (SARS-CoV-2). The aim of this project was to exploits the germicidal effect of the UVC radiation at 254 nm to sanitize the patient’s compartment of ambulances with an advanced UltraViolet SANitizing System (UV-SAN) and assess its relevance for avoiding the spread of COVID-19 and other drug resistant pathogens. Methods: The system is equipped with UVC lamps that are activated when the ambulance compartment is empty and sanitize the environment in less than 15 min. An Ozone sensor continuously monitors the gas concentration, ensuring it does not exceed threshold value harmful for patients and operators’ health. The system is relying on GNSS data and a satellite communication link, which allow to monitor and record traceability (when, where and what) of all the sanitation operations performed. This information is real-time monitored from a dedicated web-application. Results: UVC irradiation efficiently reduced SARS-CoV-2 virus titer (>99.99%), on inanimate surfaces such as plastic, stainless steel or rubber, with doses ranging from 5.5 to 24.8 mJ/cm^2^ and the UV-SAN system is effective against multi drug resistant (MDR) bacteria up to >99.99%, after 10 to 30 min of irradiation. Conclusions: UV-SAN can provide rapid, efficient and sustainable sanitization procedures of ambulances.

## 1. Introduction

Globally, ambulances perform, every year, millions of responses and their contamination with pathogenic microbes poses a serious threat to public health. The risk of infection can affect not only patients but all those who are transported inside the vehicle, first of all paramedical rescuers and medical staff, whose work involves pre or inter-hospital transfer. The safety problem is exacerbated by contamination by microbes whose pathogenesis is increased due to their recognized resistance to the main antimicrobial agents. The CoronaVirus 19 disease (COVID-19), declared a pandemic by the World Health Organization (WHO) on 11 March 2020, whose causative agent is SARS-CoV-2 virus, poses an additional danger, as the virus can be transmitted via aerosol [1,2] and can survive for hours on surfaces where deposited [3,4,5]. The COVID-19 pandemic has highlighted how sanitizations carried out correctly and repeatedly over time are a fundamental weapon to reduce the spread of pathogens in all environments and in particular healthcare ones. Contagion can be caused by contact with contaminated surfaces with SARS-CoV-2 and then touching facial areas. Not all infected people develop obvious symptoms [6]. Most infected people (81%) develop mild to moderate symptoms (up to mild pneumonia), 14% develop severe symptoms (dyspnoea, hypoxia or lung involvement greater than 50% on imaging), and 5% suffer from critical symptoms (respiratory failure, shock or multiorgan dysfunction) [7]. Especially older people or those with other co-morbidity are at greater risk of developing severe symptoms. Eliminating the risk of contagion is of fundamental importance, as the virus is able to survive on plastic and steel surfaces for up to 3 days (https://www.who.int/publications/m/item/first-data-on-stability-and-resistance-of-sars-coronavirus-compiled-by-members-of-who-laboratory-network) (accessed on 22 December 2021), depending on the temperature and light conditions [3,8]. Pathogens inactivation by physical and chemical disinfectants can leave some residual contamination [9], suggesting that a multiple disinfectant approach with efficacy UVC radiation is prudent. Effectiveness of UVC against pathogens and different strains of airborne viruses has long been established [10,11,12].

In the last decade, several studies have demonstrated the effectiveness of UVC light in the inactivation of different strains of coronavirus [13]. Since human coronaviruses have similar genomic dimension, which is the key factor in radiation sensitivity [14], it is very likely that inactivation of SARS-CoV-2 requires similar timing for the same absorbed UVC light [13]. However, the key point is to apply UVC in such a way as to sanitize as effectively as possible by eliminating the remaining pathogens on all surfaces, including the most hidden ones. Several studies reported that 49.9% of the samples collected in ambulances were positive for bacteria and that 0.9% were highly drug-resistant pathogenic strains: Methicillin-Resistant *Staphylococcus aureus* (MRSA); methicillin-resistant coagulase-negative staphylococci (MRCoNS); and carbapenemase-producing *Klebsiella pneumoniae* (KPC) [15]. In a study of 21 ambulances, 47.6% of swabs were confirmed positive for MRSA [16]. Contamination by a large number of microbes was also found in helicopter air ambulances [17], also confirmed by detailed data obtained in an Australian study [18]. Another study in four ambulances showed that “four of the seven isolated species were substantial nosocomial pathogens and three of these four possess formidable patterns of antibiotic resistance” [19]. These data are also confirmed in rural ambulances, where 49% tested positive in at least one internal point for MRSA contamination [20]. Furthermore, Gram-negative coliforms are commonly present including *Enterobacter, Klebsiella* and *Escherichia* [21]. The same study shows that the number of contaminated sites inside an ambulance increases from 57% to 86% after cleaning and disinfection. Regular and priority disinfection inside the ambulance should therefore be carried out on items frequently handled by paramedics who may be at greater risk of contamination. This type of contamination is applicable also to SARS-CoV-2 considering recent studies that have measured the persistence on surfaces [22]. Respiratory droplets and aerosol particles exhaled during the speech of infected people have a crucial role in the spread of the infection [23,24]. According to several studies virus stability and persistence on surfaces of common use such as plastic, glass, stainless steel, wood, paper, copper and cloth [3,4,25,26] seems to be influenced by the characteristics of the different materials but also by environmental conditions such as temperature, pH and humidity [4,8,27,28]. The 254 nm wavelengths of the UVC light exerts bactericidal and virucidal effects, and it is widely used in the environmental disinfection of enclosed spaces [29]. In fact, UVC light induces chemical modifications of nucleic acids, blocking the replication mechanisms [30]. Different studies demonstrated the rapid inactivation by UVC irradiation of pathogens, also of SARS-CoV-2 on droplet nuclei of bioaerosols and on surfaces and supplies [31,32,33,34,35,36,37].

During the COVID-19 pandemic period, in addition to the cleaning processes already in place, various sanitization solutions were implemented based on the use of surgical medical aids and biocidal substances. In all cases, sanitation is an operation that interrupts the ambulance service and requires the intervention of operators. In the light of the above, the efficient and rapid disinfection of surfaces may have an important role in the containment of the pandemic. In addition, the need to trace the sanitization carried out has become increasingly pressing to meet regulatory needs. In this context, an advanced UltraViolet SANitizing System (UV-SAN) was developed and integrated into an ambulance, capable of guaranteeing high levels of sanitation, and that can be used several times during the day. The system is equipped with a remote control and a traceability system exploits the germicidal effect of the UVC radiation at 254 nm to sanitize the patient’s compartment of ambulances. Through the Business Applications and Space Solutions (BASS) program (https://business.esa.int/) (accessed on 22 December 2021), dedicated to the development and validation of new Applications and Services in the downstream sector, the European Space Agency (ESA) has been active for several years in the forefront of promoting the utilization of space technologies, like Global Navigation Satellite System (GNSS), Satellite Communications (SAT-COM) or Earth Observation, and know-how to support the improvement and development of solutions in various domains, including the Health sector. The UV-SAN Project has been developed with the support of the ESA (contract reference 4000131345/20/NL/AF) in the frame of a dedicated Call for Proposals named “Space in response to COVID-19 outbreak” (https://business.esa.int/space-for-covid19) (accessed on 20 March 2021). To determine the effectiveness of the UV-SAN system, the sanitation procedure has been firstly verified on SARS-CoV-2 by testing the effect of UVC irradiation on the physiological decay of infectivity, under laboratory controlled conditions in an experimental model of disinfection on different surfaces, and secondly validated in operative environment (ambulance) by measuring its capability to reduce and inactivate multi drug resistant (MDR) bacteria.

## 2. Materials and Methods

### 2.1. System Description

UV-SAN is an UltraViolet SANitizing System designed for the sterilization of Ambulances fleets and for real-time remote monitoring of sterilization level. The system exploits the germicidal power of ultraviolet light emitted in the UVC to sterilize surfaces inside the passenger compartment of Ambulances that come into contact with the human body (e.g., hands) and which are a persistent way for the transmission of viruses and bacteria. The total germicidal effect on the vehicles, below also called Sterilization Level, is proportional to the UVC dose absorbed by surfaces, where the dose is defined as the total time-integrated amount of irradiance received. The system is equipped with four commercial low pressure mercury vapor lamps (LSE Lighting) installed on the top of the patient’s compartment in order to ensure a uniform irradiation of the compartment.

The system is designed to be used several times a day, for example during transfers when, for safety reasons, there are no patients on board, or immediately after the patient off-loading, or before a new transfer. To ensure a complete geo-traceability of the sterilization operations, UV-SAN is equipped with a Satellite Navigation System (SAT-NAV) that provides position and duration of each treatment. This information is real-time transmitted, by means of a Satellite Communication System (SAT-COM–IRIDIUM Edge^®^ with Iridium Short Burst Data Service^®^) to a Central Control System (CCS) that tracks all the sterilization operations, when and where they are performed, allowing knowing, at any time, the sterilization level of each ambulance.

The UV-SAN system is composed of two main components like shown in Figure 1.

The Sanitation Equipment (SE) is the main component of the UV-SAN system. It is installed in the rear compartment of the ambulance (Figure 2) where patients are transported and where the risk of contamination of air, surfaces and instruments is the greatest. The SE is equipped with a touchscreen that allows the healthcare operator to start the sanitation switching on the UVC lamps mounted on the ceiling of the compartment.

The position of the lamps is chosen in such a way to irradiate, as uniform as possible, the surfaces of the compartment as demonstrated by direct measure of the irradiance (i.e., the flux of radiant energy per unit area measured in W/m^2^) performed during the pilot stage.

A presence sensor ensures that the sanitation does not start if a healthcare operator, or a patient, is on board and automatically stops the irradiation whenever someone enters during the sanitation. This safety mechanism has been implemented to ensure the highest level of safety of the system with the scope to reduce at maximum the risk of accidental exposure to UV radiation. As noted, the UVC radiation stimulates the production of ozone (O_3_) starting from molecular oxygen of the atmosphere [38]. O_3_ is highly corrosive to equipment and is lethal to humans with prolonged exposure at concentrations above 4 ppm. The U.S. Food and Drug Administration (FDA)’s maximum allowed ozone concentration in the air for residential areas is 0.05 ppm ozone by volume. The SE is equipped with an O_3_ and UVC sensors (respectively Spec-Sensor DSG-03 and D-SICONORM by Z-E-D Electronics) that measure the gas concentration and the UV irradiance level avoiding they exceed the safety level defined by the international normative.

The CCS is the component that records all the operations carried out by each SE in order to maintain full traceability of the operations performed and the sterilization level of each ambulance. The CCS can also update configuration parameters in order to guarantee the same performances to all the equipment installed. Data collected can be visualized by a Web-Application (https://www.uvsan.org) (accessed on 22 December 2021) that allows defining configuration parameters of each device, monitors the status of sanitation level, checks when and where sanitations have been performed and other relevant statistics. Finally, all the data can be filtered and downloaded for further analysis and for inspection scope.

Once the trial unit was deployed and successfully tested, the adequacy of system with respect to the End User needs has been evaluated from an operational point of view. A set of six Key Performance Indicators (KPIs) covering both technical issues, like the real-time monitoring or the reliability of safety mechanisms based on measurement of ozone concentration and human presence, and operational aspects, like the ease of use or the reduction of workload effort, have been evaluated.

Furthermore the new sanitation procedure implemented by the system was compared with that ones used during the pandemic period to assess benefits and advantages brought. The analysis is made with respect to operative parameters like, the effort and time needed or service down-time, and administrative parameters like the provision of traceability.

### 2.2. Cells

Vero-E6 (*Cercopithecus aethiops* derived epithelial kidney, C1008 ATCC CRL-1586) cells were grown in Dulbecco’s Modified Eagles (DMEM) High glucose 4.5 g/L (Gibco, Life Technologies Italia, Monza, Italy) supplemented with 10% fetal calf serum (Corning, Mediatech Inc., Manassas, VA, USA) and 100 units/mL penicillin/streptomycin (Gibco), 100 units/mL penicillin, 100 μg/mL streptomycin, 2 mM Lglutamine, 1 mM sodium pyruvate (Gibco), and 1× non-essential amino acids (Gibco). All cells were grown at 37 °C in a 5% CO_2_ humidified incubator.

### 2.3. SARS-CoV-2 Strain

All experiments were performed using the strain BetaCov/Italy/CDG1/2020|EPI ISL 412973|2020-02-20 (GISAID accession ID: EPI_ISL_412973). The virus was isolated in Mila, Italy, and propagated as previously described [39]. Briefly, the viral isolate was propagated by inoculation of 70% confluent Vero-E6 cells in 75 cm^2^ cell culture flasks. Stocks of SARS-CoV-2 virus were harvested at 72 h post infection, and supernatants were collected, clarified, aliquoted, and stored at −80 °C. The virus was quantified by end-point titration, employing the Spearman-Karber method [40,41], on Vero cells by TCID_50_ (dilution of virus at which 50% of the cell cultures are infected), with a titer of 10^5.76^ TCID_50_/mL. All viral manipulations were conducted within biosafety level (BSL)-3 facilities at Istituto Superiore di Sanità (ISS, Rome, Italy).

### 2.4. Inactivation of SARS-CoV-2 by UVC-Irradiation

SARS-CoV-2 susceptibility to 254 nm UVC treatment was tested on plastic (polystyrene), metal or rubber surface. Viral suspensions (30 μL) containing ca. 1.73 × 10^4^ TCID_50_, were loaded onto a sterile polystyrene plate (Corning Incorporated-Life Sciences, Oneonta, NY, USA), on rubber or metal sterilized dishes (9.2 cm^2^) and spread into a circle. The inoculum was allowed to dry under sterile laminar flow in a biosafety cabinet at room temperature for 30 min. The stabilized deuterium UV light source used of the experiment emits light with a wide wavelength spectrum (Thorlabs LTD., Ely, UK) as reported in Figure 3. The specific wavelength of 254 nm, the same emitted by the UV-C lamps used in the ambulance, is selected using a narrow band filter centered at 254 ± 2 nm.

The irradiance of the deuterium lamp at 254 nm has been measured with a calibrated silicon carbide diode (SiC) UVC sensor (Reference Sensor D-SiCONORM-LP-REF) placed at 3 cm from the UV source. The distance of 3 cm corresponds to the distance between the filter and the support plate (polystyrene, metal or rubber) where the virus is loaded and dried. The value of irradiance measured by the UVC sensor was 10 μW/cm^2^ corresponding to 0.01 mJ/cm^2^ per second.

The effect of 254 nm UVC irradiation on the virus was tested in three independent experiments, each performed in quadruplicate up to 60 min of exposure for polystyrene, rubber and metal surfaces. The samples were taken resuspending the dried virus in 300 μL of DMEM 2% FCS after 0, 10, 20, 30, 40 and 60 min of exposure that correspond to an UVC dose of 0, 6, 12, 18, 24, 36 mJ/cm^2^, respectively). Each experiment included not-irradiated virus as control. All experiments with the SARS-CoV-2 virus were performed in a biosafety level 3 (BLS-3) containment facility at ISS.

### 2.5. SARS-CoV-2 Viral Titration

To determine the TCID_50_ of SARS-CoV-2 virus UVC-exposed, collected samples were serially diluted, and used to inoculate Vero E6 cells. To this purpose, 96-well plates (Corning, Mediatech Inc., Manassas, VA, USA) were inoculated with 2-fold dilutions in quadrupled of each viral sample (100 μL/well), and 22,000 cells/well were added. Cells were then incubated for 5–6 days and were checked daily to observe the cytopathic effect. TCID_50_/mL was calculated according to the Spearmane-Karber method. Dose UVC efficacy of the decontamination process was indicated as Log reduction and percent reduction of viral titer.

### 2.6. UVC Disinfection Test on Ambulance

The effectiveness of the UV-SAN sanitation procedure in the operating environment was tested against Methicillin Resistant *Staphylococcus aureus* (MRSA) and Extended-Spectrum β-Lactamase (ESBL) *Klebsiella pneumonia*, common pathogens wide-spread in the hospital environment. One hundred μL of bacterial cultures suspension at different concentration (at 0.5, 1.0 and 2 McF), was distributed by drops on different parts of the ambulance surface. After contamination, the UV-SAN system was applied as indicated by the manufacturer’s instruction, by specific lighting cycles lasting 10, 20 and 30 min. In order to have a calibrated measure of the UVC dose absorbed, a measure of the irradiance in the sample area was performed using the same calibrated sensor D-SiCONORM-LP-REF used for calibrating the laboratory experiments.

After the sanitation procedure, samples from surface swabs have been collected and inoculated on specific bacterial selective media as already described [42]. Inoculated agar plates were incubated for 24 and 48 h at 35 °C. After incubation plates were observed for bacterial growth. Any bacterial colony found was identified by MALDI-TOF and antimicrobial susceptibility testing, as already described [43]. The experimental phase was performed at the Laboratory Unit of the University Campus Bio-Medico of Rome.

## 3. Results

### 3.1. Evaluation of 254-nm Ultraviolet Light on Disinfecting SARS-CoV-2 Surface Contamination

The effectiveness of the UVC radiation against SARS-CoV-2 has been tested in a biosafety level 3 (BLS-3) containment facility at the ISS. About 1.7 × 10^4^ TCID_50_ were placed upon each material surface with a sterile pipet tip and let it dried. SARS-CoV-2 showed high susceptibility to UVC irradiation on different surfaces as reported in the log-scale graph of Figure 4 where the viral load is reported as function of the exposure time (min). On polystyrene surface, an UVC dose of 24 mJ/cm^2^ of 254 nm UVC light (0.01 mJ/cm^2^ per second, for 40 min) reduced SARS-CoV-2 viral load to undetectable levels, measured by calculation of 50% endpoint by serial dilution (TCID_50_). On a metal surface, 18 mJ/cm^2^ of UVC light (0.01 mJ/cm^2^ per second, for 30 min) was enough to reduce the viral load to undetectable levels, while disinfection of the rubber only required 3 mJ/cm^2^ (0.01 mJ/cm^2^ per second, for 5 min).

Experimental data for the three materials was fitted using a linear regression model. The goodness of the fit is measured by the R^2^ value that is equal to 0.947 (*p*-value: 5.7 × 10^−3^) and to 0.944 (*p*-value: 5.3 × 10^−3^) respectively for metal and polystyrene. A lower value of R^2^ is measured for the rubber 0.66 (*p*-value: 9.3 × 10^−2^). Results are summarized in Table 1, reporting the 95% confidence interval for obtaining the log3 reduction. The SARS-CoV-2 TCID_50_/mL values reported in Figure 4 are normalized with respect to the control measurements of the sample carried out at the same time but without exposure to UVC radiation. The control measure was used to evaluate the percentage of reduction in viral concentration due to environmental factors. After 40 min, the reduction of the TCID_50_/mL for polystyrene was of 22%, while on metal and rubber were of 10% and 6% respectively, demonstrating that the viral load on the surfaces slowly reduces over time.

From the value of the regression model we evaluated the UVC dose need to achieve 90% to 99.99% of virus infectivity reduction on different surfaces (log1 to log4) virus infectivity, as reported in Table 2.

### 3.2. On Site Effectiveness of UV-SAN System

The effectiveness of the UV-SAN sanitation procedure in the operating environment was tested against the multi drug resistant (MDR) bacteria Methicillin resistant *Staphylococcus aureus* (MRSA) and Extended-Spectrum β-Lactamase (ESBL) *Klebsiella pneumonia*, common pathogens wide-spread in the hospital environment. Contamination was focused especially in the part that are presumably closer or in direct contact with the patient during transport, as the headrest area, the arm-rest area, the chair and the area of the ambulance located near the urinary catheter tube passage or where the drainage bag lies in case the patient is carrying a urinary catheter, as shown in Figure 5.

The UV-SAN treatment for 10 to 30 min of the ambulance resulted in a significant reduction of bacteria growth, as indicated in Table 3.

Any bacterial growth was detected at the headrest, armrest and chair surfaces, after contamination by increasing concentrations of bacterial suspension (0.5, 1.0 and 2.0 McF) at different exposure times (10, 20 and 30 min). In the area where the urinary catheter tube could presumably be located, in the event that the patient has it, area located in a recess on the side of the stretcher, a significant growth of MRSA has been detected after UV-SAN application. Exactly, 2 CFU/mL was recorded at the concentration of 0.5 McF after exposure for 10, 20 and 30 min, while at increasing concentrations (1.0 and 2.0 McF) the growth was higher and >100 CFU/mL at the exposure time of 10 min. To obtain a significant decrease in bacterial growth at any of the concentrations used it was necessary to apply a UV-SAN for at least 20 min. Anyway, exposure up to 30 min did not ensure the absence of bacterial growth even at low concentrations (0.5 McF).

### 3.3. Trial Phase

The trial phase started in Rome (Italy) in June 2021 after the installation of the trial unit. Figure 6 shows the system while performing a sanitation.

The data collected by the sensors on board and registered after a sanitation (i.e., time, position, duration, level of sanitation reached) are real-time transmitted to the CCS and available to the User by a dashboard like that one shown in Figure 7. The user can verify information like the number of sanitation day-by-day, the last sanitations performed and check how many sanitations have been performed for each ambulance.

On the basis of the data collected and from feedback collected by the Pilot User the assessment of the KPIs demonstrated that the system achieved the expected results both considering the technical issues and the operative aspects as reported in the Table 4.

A further analysis was performed comparing the cleaning and sanitation procedure used during the 2020 pandemic period and the current implemented with that one proposed with the UV-SAN. This assessment is mainly focused on the operational point of view highlighting the benefits that this type of system could bring. The outcomes are reported in Table 5.

## 4. Discussion

Ambulance vehicles are an integral part of emergency medical services. They can respond to thousands of cases per year and after each mission, each ambulance vehicle must be cleaned and decontaminated to be ready for use for the next mission. The possibility of having increasingly effective cleaning protocols for these environments is essential to limit the presence of pathogenic microorganisms that represent a potential risk of infection for patients and paramedics. Furthermore, an ineffective disinfection of the ambulance and the contamination of patients or transported personnel can contribute to the spread of pathogenic microorganisms in hospital environments and thus contribute to the development of health care. A recent study has shown that the greatest reduction in the risk of acquiring pathogen infections in a hospital setting can be achieved with the use of enhanced environmental disinfection strategies, especially with the addition of a UV-C device to the standard disinfectant strategy [44]. In the present study, an advanced UltraViolet SANitizing System (UV-SAN) has been developed to disinfect ambulance vehicles from dangerous pathogens, like SARS-CoV-2 but also from MDR bacteria.

Based on our in vitro results, a smaller dose of UVC (55.8–248 mJ/m^2^) is enough to reduce the SARS-CoV2 viral titer of >99.99% under experimental conditions. In particular, plastic seems to be the most refractory material to UVC disinfection, followed by stainless steel and rubber, where the latter showed the best compliance with the treatment. These results are comparable to those obtained from previous studies where laboratory tests have shown that the UVC susceptibility of virus depends on the type of surface [22,31,33].

To evaluate the irradiation times of the UV-SAN system inside the ambulance a calibrated mapping of the irradiance was performed inside the vehicle. This mapping was carried out measuring the irradiance in different points with the same calibrated Reference Sensor D-SiCONORM-LP-REF, as reported in Table 6.

Once the UVC dose necessary to obtain a log reduction of the virus, *(D_log_n_)*, is known and the irradiance (*Irr)* in a certain point measured, the irradiation time, (*t_exp_log_n_)*, is calculated with the following equation:*t*_*exp_log_n*_ = *D*_*log_n*_/*Irr*(1)

By applying Equation (1), the time required to obtain a reduction of up to 99.99% was calculated for several points within the ambulance, also taking into account the material, as reported in Table 6.

The UV-SAN system could be able to achieve a viral reduction equivalent to 99.99% by after about 10 min of irradiation. The irradiation inside the ambulance and the exposure to UVC radiation of the surfaces clearly depends on the arrangement of the source lights which, to obtain a more uniform irradiation, must be positioned on the ceiling, illuminating from the top towards the low.

Measurements carried out with the MDR bacteria also showed that a 10 min irradiation is effective against *Klebsiella pnemumoniae ESBL* even in the less exposed points, while 20 min are needed to significantly reduce the presence of *Staphylococcus aureus MRSA* in the less exposed areas. In this regard, it is important to note how the system can be used several times a day in order to have a cumulative effect of exposure, further reducing the presence of the pathogens agents.

Test performed in the ambulance confirms two relevant issue related to inactivation of bacteria using UVC radiation:

UVC exposure greater than 20 mJ/cm^2^ are needed to observe a significant reduction both of *Klebsiella pneumoniae ESBL* and *Staphylococcus aureus MRSA* as reported in previous experiment [45,46].

Greater resistance of the *Staphylococcus aureus* was measured confirming the already observed evidence that Gram-positive cells tend to be more intrinsically resistant to UV exposure [47,48].

For this reason a more prolonged treatment (e.g., 30 min) is advisable to eliminate all bacterial contaminating the different ambulance areas even those more hidden. It is important to highlight that the UV-SAN system has been designed to be used many times a day to have a cumulative effect of exposure, further reducing the presence of the pathogens agents. For example, the activation for 10 min, a time that does not cause a significant interruption of the service, for six times during a shift brings to a total exposure of 1 h that is enough to inactivate pathogens also in more hidden areas.

## 5. Conclusions

UV-SAN is a sanitation system based on UVC radiation that can be installed inside ambulances and that is able, in a short time, to carry out an effective, safe and sustainable sanitation of the patient’s compartment.

The effectiveness of the UVC radiation has been successfully tested both against SARS-CoV-2, and MDR bacteria. Experimental tests have shown that UVC radiation is effective in 10 min to reduce SARS-CoV-2 viral load to undetectable level, moreover the UVC light can be efficiently implemented in the UV-SAN system for the sanitation of ambulance compartments where it was validated against MDR bacteria. Due to the precautions required by the Biosafety Level (BSL-3) for testing a pathogen, such as SARS-CoV-2, it is a standard technique to use surrogate species as a reference to higher BSL species, including bacteria [49,50]. Thus, it could be assume that the efficacy results obtained on the ambulance with MDR bacteria, could be extrapolated to the SARS-CoV-2.

The ease of use and the short irradiation time is compatible with the daily routine apply use even during periodic downtime (i.e., patient transfer, waiting after patient off-load). These aspects were also verified during a pilot stage directly on the ambulance started in June 2021. The daily use has shown that the system does not need additional efforts by the operators. The automatic recording of the sanitation process allows the Ambulance Manager to have a real-time and full control of sanitation level and to retrieve this information, needed to accomplish administrative and legal duties, in a short time and with a high level of confidence. In fact, the traceability process is fully automatized without the intervention of the healthcare operator, guaranteeing the truthfulness of the data.

These data are encouraging and indicate that UV-SAN can be used as a complementary sanitization system in addition to the current cleaning methods to increase the level of sanitation, to prevent the accidental spread of pathogens and to reduce the risk of an infections.

The threatening spread of MDR bacteria inside hospitals urges us to look for new solutions, in order to minimize the risk for patients and health care workers, thus although the UV-SAN System has been designed and optimized to work in ambulances, considering the time required for sanitation and ease of use, it can find applications in other healthcare environment.

## Figures and Tables

**Figure 1 ijerph-19-00331-f001:**
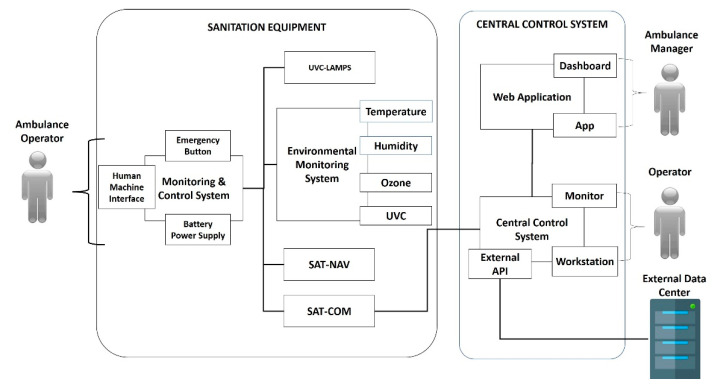
UV-SAN System Architecture.

**Figure 2 ijerph-19-00331-f002:**
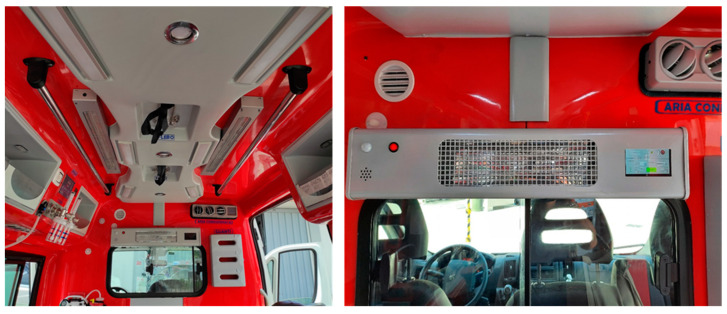
UV-SAN Sanitation Equipment installed in the ambulance patient compartment.

**Figure 3 ijerph-19-00331-f003:**
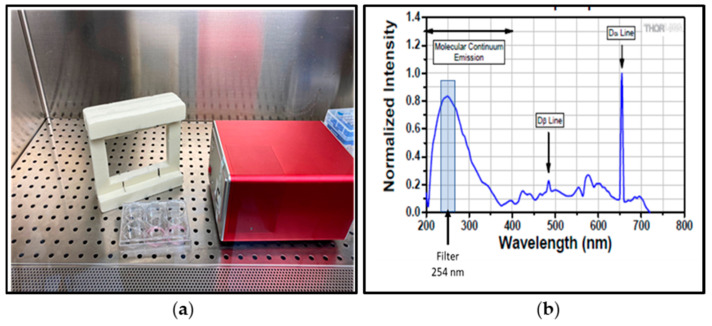
Setup of the experiments: (**a**) SARS-CoV-2 dried on polystyrene surface before treatment with UVC. (**b**) Emission spectrum of the stabilized deuterium lamp used for the experiments. The blue-shaded region indicates the lamp’s operating wavelength range (modified from https://www.thorlabs.com) (accessed on 22 December 2021).

**Figure 4 ijerph-19-00331-f004:**
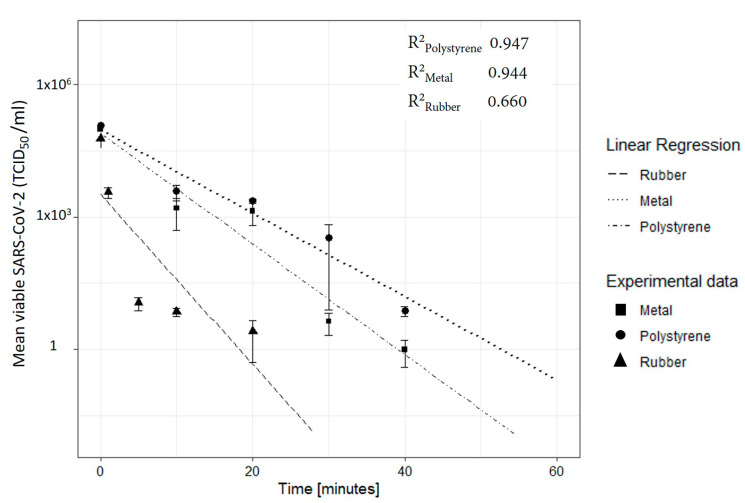
UVC Susceptibility measured on different surfaces. The mean ± SEM values from experiments (*n* = 3) are indicated for polystyrene, metal and rubber; the lines represent the best-fit linear regressions of experimental data.

**Figure 5 ijerph-19-00331-f005:**
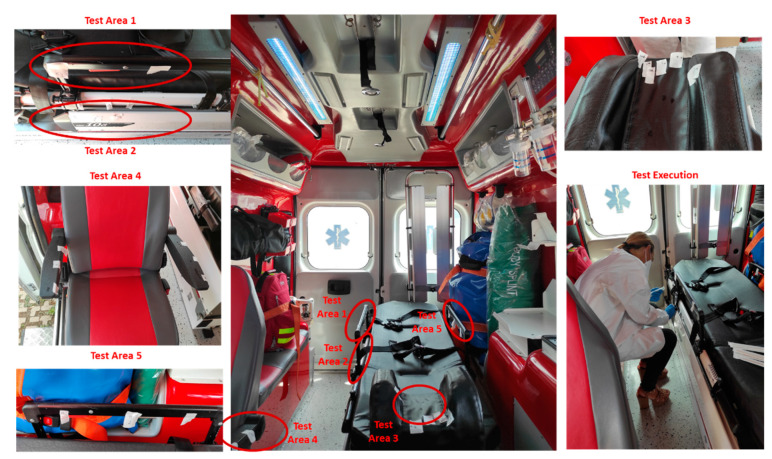
Test Execution on Ambulance during the Pilot phase.

**Figure 6 ijerph-19-00331-f006:**
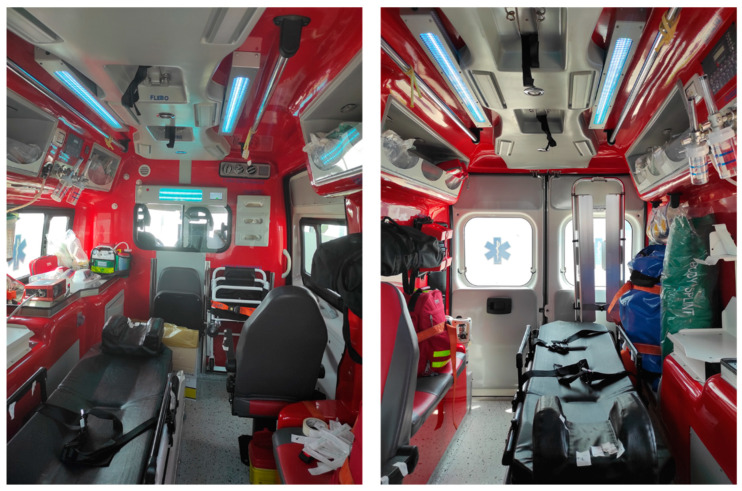
UV-SAN system deployed.

**Figure 7 ijerph-19-00331-f007:**
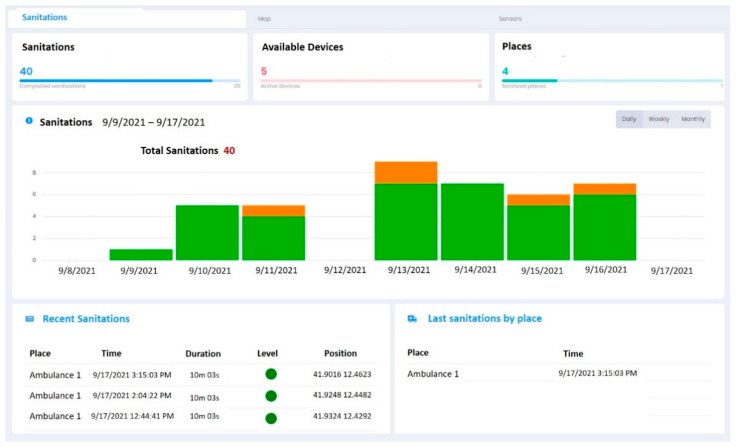
UV-SAN—Dashboard for consultation of sanitation data.

**Table 1 ijerph-19-00331-t001:** Summary of Linear Regression Model.

Material	UVC Dose for Log3 Reduction [J/m^2^] 95% Confidence Level			*p* Value	R^2^
	Lower	Mean	Upper		
Polystyrene	135	184	226	5.3 × 10^−3^	0.944
Metal	92	129	164	5.7 × 10^−3^	0.947
Rubber	29.9	54.4	98.3	9.3 × 10^−2^	0.66

**Table 2 ijerph-19-00331-t002:** UVC dose for different materials and % of virus infectivity reduction.

Log Reduction	% Reduction	UVC Dose [J/m^2^]		
		Polystyrene	Metal	Rubber
Log1	90%	57.3	46.2	7.8
Log2	99%	121	87.5	23.4
Log3	99.9%	184	129	54.4
Log4	99.99%	248	170	55.8

**Table 3 ijerph-19-00331-t003:** Results of UVC susceptibility test on MDR bacteria performed in the ambulance.

Position	Pathogen and Concentration	Exposure Time [min]	Irradiation [mJ/cm^2^]	Growth after Incubation	
				24 h	48 h
HeadRest	*Kleb. Pne. ESBL+* *0.5, 1, 2 McF **	10	76	None	None
		20	152	None	None
		30	228	None	None
	*Staph. aur. MRSA* *0.5, 1, 2 McF **	10	76	None	None
		20	152	None	None
		30	228	None	None
Armrest	*Kleb. Pne. ESBL+* *0.5, 1, 2 McF **	10	44	None	None
		20	88	None	None
		30	132	None	None
	*Staph. aur. MRSA* *0.5, 1, 2 McF **	10	44	None	None
		20	88	None	None
		30	132	None	None
Chair	*Kleb. Pne. ESBL+* *0.5, 1, 2 McF **	10	45	None	None
		20	90	None	None
		30	135	None	None
	*Staph. aur. MRSA* *0.5, 1, 2 McF **	10	45	None	None
		20	90	None	None
		30	135	None	None
Urinary catheter zone	*Kleb. Pne. ESBL+* *0.5, 1, 2 McF **	10	18	None	None
		20	36	None	None
		30	54	None	None
	*Staph. aur. MRSA* *0.5 McF*	10	18	2	2
		20	36	2	2
		30	54	2	2
	*Staph. aur. MRSA* *1 McF*	10	18	>100	>100
		20	36	3	3
		30	54	2	2
	*Staph. aur. MRSA* *2 McF*	10	18	>100	>100
		20	36	3	3
		30	54	2	2

* Test repeated for three different concentration with same result.

**Table 4 ijerph-19-00331-t004:** Key Performance Indicators evaluation.

KPI	Definition	Type of Measure	Expected Result	Result
KPI 1	SARS-CoV-2 and MDR pathogen inactivation	Laboratory Test On-Site Test	Reduction >99%	Confirmed in about 10 min
KPI 2	Monitoring & Traceability	Analysis of DB data	<1% loss of data	<0.1%
KPI 3	Real Time Monitoring	Analysis of DB data transmitted with standard comms link	<1 min	<30 s
KPI 4	SAT-COM data transmission	Analysis of DB data of data transmitted with SAT-COM	<3 min	<150 s
KPI 5	Safety Check (ozone, human presence)	Reports from Pilot User	<1%	No problem detected
KPI6	Time & Work-load Reduction	Report from Pilot User	No downtime of ambulance service	Confirmed

**Table 5 ijerph-19-00331-t005:** Comparison between sanitation procedures.

Sanitation Procedure for Transportation of COVID-19 Patients			
	Pandemic Period 2020	Pandemic Period 2021	UV-SAN
Procedure	Sanitation performed after the transportation of each COVID-19 patient: Patient transported Ambulance to the depot (unavailable for other transport): 30 min Sanitation: 1 h (even more during most critical period because of accumulating ambulances) Ambulance to the hospital (available for new transport): 30 min	Sanitation performed only at the end of the shift: Ambulance to the depot (unavailable for other transport): 30 min Sanitation: 1 h (even more during most critical period because of accumulating ambulances)	Sanitation performed at any time when the compartment is empty
Time needed for the sanitation	2 h		Time needed for the sanitation
Service down-time	2 h for each transport	None (sanitation performed at the end of the shift)	None
Location	Depot	Depot	Everywhere
Staff needed	1 person fully employed	1 person fully employed	1 person partially employed (start and stop sanitation)
Traceability	Manually recorded	Manually recorded	Automatic

**Table 6 ijerph-19-00331-t006:** Evaluation of time needed to reduce the SARS-CoV-2 virus over different surfaces in the patient compartment of the ambulance.

Surface Characteristics			Time for Log Reduction [s]			
Position	Material	Irradiance [W/cm^2^]	Log1	Log2	Log3	Log4
Instruments shelf	Plastic	1	46	88	129	170
Stretcher base	Metal	0.45	127	269	410	551
Back Door	Plastic	0.24	193	365	537	709
Lateral seat	Plastic	1	46	88	129	170
Seat	Plastic	0.35	132	250	368	486
Floor 1	Rubber	0.65	12	36	84	132
Headrest	Plastic	1.27	36	69	101	134
Floor 2	Rubber	0.65	12	36	84	132
Stretcher Armrest (right)	Metal	0.74	77	163	249	335
Stretcher Armrest (left)	Metal	0.72	80	168	256	344
Seat Armrest (right)	Plastic	0.75	62	117	172	227
Seat Armrest (left)	Plastic	1.34	34	65	96	127

## Data Availability

Not applicable.

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
