# Peer review of "UltraViolet SANitizing System for Sterilization of Ambulances Fleets and for Real-Time Monitoring of Their Sterilization Level"

_ijerph, 2021, doi:10.3390/ijerph19010331_

Round 1

Reviewer 1 Report

The article presents the results of interdisciplinary applied research on the development of a comprehensive system for disinfecting the equipment of ambulances.
The system being the subject of this article covers the functions of disinfection with the use of UVC radiation as a biocidal agent and an advanced subsystem for the acquisition, archiving and data transfer of disinfection operations.
Research design is constructed correctly in terms of methodology and includes the necessary types of research.
While it is expressed in terms of efficacy against Sars-Cov-2 virus, efficacy studies against this agent were performed (for obvious reasons) under laboratory conditions, and on-site studies were performed only on bacteria.
The results of the latter, of course, can be extrapolated to viruses, due to their lower resistance than bacteria having their defense mechanisms. This, however, should be clearly emphasized.
The novelty is not an asset to this article. The use of UVC as a biocidal agent is known, as the authors note in the introduction. However, studies on the effectiveness of real virus biocides under BSL3 conditions are rare and are very valuable due to both the scientific and utilitarian aspects.
The results are presented and reported clearly, however, it would be advisable to introduce some elements of the statistical evaluation, considering that the obtained parameterized results charged with an objective error will be used to estimate the derivative applicable parameters as eg. the exposure time (equation 2).
Although the scientific importance of the subject of the article is not the main advantage, its significance of content in the practical aspects of dealing with infections of nasocomial origin, in particular in the aspect of Covid occurs to be difficult to overestimate. 
In the article I miss a reference to other methods of disinfecting ambulances (e.g. peroxide fumigation) as well as to the limited access of UV light to potential outbreaks of bacteria and viruses.
It would also be worth mentioning the potential biocidal properties of UVC against virions present on droplett cuclei of bioaerosols, which are the key infectious agent.
I consider that the article deserves to be published in the journal. 

Before submitting it for publication, please take note of the following comments.
1. There are many abbreviations in the text. A collective explanation, e.g. at the end of the text, would be valuable.
2. Pls. introduce elements of the statistical evaluation of the obtained results.
3. A lot of "Reference not found" warnings to be corrected.
4. Row 133: "(...) positioned on the roof (...)" - maybe "(...)mounted on the ceiling(...)"?
5. Pls use greek character "μ" instead of "u" where necesary (eg. row 207: "10 uW / cm2")
6. Correct the proper character cases where necessary (see example above)
7. Row 228: pls correct "be-ta lactamase" to "β-Lactamase"
8. Row 249: improper dimension of irradiance: is "24 mJ / cm2" - should be "24 mW / cm2"
9. Pls use italics for Latin names of species
10. Pls consider unification of radiometric terms (not easy task ...). From my point of view the "exposure dose" in tab. 1. and "irradiation", except of the obvious same dimensions have rather same physical meaning; in row 208 "light dose"...
11. Unfortunately, the screen shown in Fig. 7. is in Italian and therefore may be illegible to many readers 

Reviewer 2 Report

This paper tried to use the Ultraviolet Sanitizing System for sterilization of Ambulances 2 fleets and for real-time monitoring of their sterilization level. Here are some comments for the authors consideration:

  1. The innovative points and scientific contributions were low. Many existing studies used UV-C disinfection, so what is the unique innovation of this study?
  2. There are many similar products in the market, what is the unique of the UV-SAN Sanitation Equipment developed in this study?
  3. The organization of the whole paper was confusing and misleading.
  4. The quality of the Figures should be improved, for example Figure 4.
  5. Figure 7 was not suitable in the scientific research articles.

Reviewer 3 Report

The paper is scientifically based.

Sometimes the paper is similar to an application for an invention, which obviously it is not.

I think the paper can be accepted for publication after correction.

Specific comments

Line 126 

Is this figure is original? If it is borrowed, then it needs to provide a link/reference.

What does the “Space technology” in the figure mean?

Whether it is necessary to decipher/explain some of the abbreviations in the figure (“HMI” - human-machine interface and etc)? - 

Line 200

both pictures in figure3 are not original - / is it necessary to provide a link to the original pictures?

Line 340

Table 5

I think it is necessary to provide experimental data on the dynamic of the survival of the virus under the same conditions and on the same surfaces without exposure to ultraviolet.

Line 375-576 Conclusion

”UV-SAN is the first sanitation system based on UVC radiation that can be installed 375 inside ambulances and that is able, in a short time, to carry out an effective, safe and sustainable sanitation of the patient's compartment.”

I doubt this statement - it does not follow from the text of the paper and rather looks like a technical text from an application for inventions.

As far as I understand, the authors propose a complementary sanitization system in addition to the current cleaning methods. That is, any standard UV lamp - which is found in ambulances - will work just as effectively if they are periodically turned on during the working day.

Reviewer 4 Report

The manuscript sounds interesting and can be accepted with minor revision. Some of my comments are below.

  1. The introduction needs a bit more information about the effects of SARS COVID19. Please use the reference below.

https://www.nature.com/articles/s41598-021-85425-w

https://www.sciencedirect.com/science/article/pii/S2590137021000121

2.  How does the author justify the COVID surviving in the surfaces of the ambulance? Do we have a control to prove that virus survives in the surface?
